# In Situ Epitaxial Quantum Dot Passivation Enables Highly Efficient and Stable Perovskite Solar Cells

**DOI:** 10.3390/nano15130978

**Published:** 2025-06-24

**Authors:** Yahya A. Alzahrani, Raghad M. Alqahtani, Raghad A. Alqarni, Jenan R. Alnakhli, Shahad A. Anezi, Ibtisam S. Almalki, Ghazal S. Yafi, Sultan M. Alenzi, Abdulaziz Aljuwayr, Abdulmalik M. Alessa, Huda Alkhaldi, Anwar Q. Alanazi, Masaud Almalki, Masfer H. Alkahtani

**Affiliations:** 1Future Energy Technologies Institute, King Abdulaziz City for Science and Technology (KACST), Riyadh 11442, Saudi Arabia; yalzhrani@kacst.gov.sa (Y.A.A.); aalessa@kacst.gov.sa (A.M.A.); aqalanazi@kacst.edu.sa (A.Q.A.); mhmalki@kacst.gov.sa (M.A.); 2Department of Physics, College of Science and Humanities, Imam Abdulrahman Bin Faisal University, P.O. Box 1982, Jubail 31441, Saudi Arabia2210003683@iau.edu.sa (S.A.A.); 3Department of Chemistry, King Saud University, P.O. Box 2455, Riyadh 11451, Saudi Arabia

**Keywords:** perovskite quantum dots (PQDs), core–shell structure, epitaxial passivation, perovskite solar cells (PSCs), photovoltaic performance enhancement

## Abstract

We report an advanced passivation strategy for perovskite solar cells (PSCs) by introducing core–shell structured perovskite quantum dots (PQDs), composed of methylammonium lead bromide (MAPbBr_3_) cores and tetraoctylammonium lead bromide (tetra-OAPbBr_3_) shells, during the antisolvent-assisted crystallization step. The epitaxial compatibility between the PQDs and the host perovskite matrix enables effective passivation of grain boundaries and surface defects, thereby suppressing non-radiative recombination and facilitating more efficient charge transport. At an optimal PQD concentration of 15 mg/mL, the modified PSCs demonstrated a remarkable increase in power conversion efficiency (PCE) from 19.2% to 22.85%. This enhancement is accompanied by improved device metrics, including a rise in open-circuit voltage (Voc) from 1.120 V to 1.137 V, short-circuit current density (Jsc) from 24.5 mA/cm^2^ to 26.1 mA/cm^2^, and fill factor (FF) from 70.1% to 77%. Spectral response analysis via incident photon-to-current efficiency (IPCE) revealed enhanced photoresponse in the 400–750 nm wavelength range. Additionally, long-term stability assessments showed that PQD-passivated devices retained more than 92% of their initial PCE after 900 h under ambient conditions, outperforming control devices which retained ~80%. These findings underscore the potential of in situ integrated PQDs as a scalable and effective passivation strategy for next-generation high-efficiency and stable perovskite photovoltaics.

## 1. Introduction

Organic–inorganic hybrid halide perovskites have emerged as revolutionary materials in the field of optoelectronics due to their exceptional optoelectronic properties, including high absorption coefficients, long carrier diffusion lengths, tunable bandgaps, and low-cost, solution-based processability [1,2,3,4,5]. Among their most prominent applications, perovskite solar cells (PSCs) have demonstrated remarkable progress, reaching certified power conversion efficiencies (PCEs) exceeding 27% within just over a decade [6,7]. These advancements position PSCs as strong alternatives to conventional silicon-based photovoltaics for both single-junction and tandem configurations [1,4,6,7]. Despite their outstanding performance metrics, the practical deployment of PSCs is severely hindered by their intrinsic instability under environmental stressors such as moisture, oxygen, ultraviolet (UV) light, and thermal cycling [2,4,8]. The degradation mechanisms are primarily rooted in structural instabilities and the formation of defects at grain boundaries and interfaces, which facilitate ion migration, accelerate non-radiative recombination, and ultimately compromise device performance and operational lifetime [3,4,5,9]. Therefore, developing effective passivation strategies to mitigate these defects and stabilize device interfaces is of paramount importance.

Recent efforts have increasingly focused on integrating nanomaterials into PSC architectures to address these challenges. Metal oxide nanoparticles have been extensively utilized to enhance charge extraction in transport layers [10,11,12,13], while luminescent nanostructures such as carbon dots [14,15], graphene derivatives [16,17], and lanthanide-doped upconversion (UCNPs) [18,19,20] and downconversion (DCNPs) nanoparticles [21,22,23] have provided additional functionalities for optical management and defect passivation. In particular, spectral converters like UCNPs and DCNPs have shown promise in simultaneously improving light utilization and mitigating UV-induced degradation. UCNPs absorb near-infrared (NIR) photons and re-emit in the visible spectrum, extending the absorption range of PSCs [18,20], whereas DCNPs convert high-energy UV photons into visible light, protecting sensitive layers such as TiO_2_ from photodegradation [23]. However, these nanostructures face critical limitations when implemented in PSCs. For instance, the upconversion efficiency of UCNPs remains inherently low under the weak photon flux of solar illumination, significantly limiting their contribution to photocurrent generation. Additionally, their insulating matrices, suboptimal film compatibility, large particle sizes, and poor dispersion characteristics often interfere with charge transport and film uniformity, posing challenges for large-scale fabrication and device reproducibility [8,9]. To develop more chemically compatible and stable passivation strategies, recent studies have turned toward perovskite quantum dots (PQDs), which share compositional similarity with bulk perovskites and offer favorable energy level alignment. In earlier studies, PQDs were deposited onto perovskite films via spin-coating or drop-casting [24,25,26]. These approaches aim to reduce surface defect densities and enhance charge transport by minimizing trap-assisted recombination and improving interfacial energy alignment. Although such techniques have led to modest increases in PCE and film uniformity, PQDs formed ex situ are often unstable under ambient conditions due to their high surface-to-volume ratio and lack of encapsulation [6,7,26]. Moreover, these PQDs are not chemically bonded or epitaxially matched to the host perovskite lattice, leading to weak interfacial adhesion and limited long-term passivation efficacy.

Building upon the emerging concept of quantum dot-enhanced passivation, a more advanced and effective strategy involves the in situ epitaxial growth of PQDs during the crystallization of the perovskite film [27,28]. During the nucleation and growth of the perovskite layer, PQDs spontaneously form and become embedded at grain boundaries and surfaces, benefiting from favorable lattice matching and strong interfacial bonding with the host matrix. This in situ integration strategy has been shown to enhance both the power conversion efficiency (PCE) and operational stability of perovskite solar cells. However, the unpassivated surfaces of the PQDs remain chemically active and susceptible to environmental factors such as moisture and oxygen, leading to degradation over time and compromising long-term device performance [29].

To address this challenge, core–shell quantum dot architectures, widely used in colloidal nanocrystal systems, have garnered increasing interest [30,31]. These structures, in which a low-bandgap core is encapsulated by a higher-bandgap shell, effectively suppress non-radiative surface recombination, enhance carrier confinement, and improve chemical and thermal robustness. Inspired by this paradigm, researchers have begun exploring the integration of core–shell PQDs into many photonics applications [30,32,33] and have shown a better performance; however, their use in PSCs remained unexplored.

In this study, we present a dual-functional approach that synergistically combines core–shell PQDs design with in situ epitaxial incorporation. Specifically, MAPbBr_3_@tetra-OAPbBr_3_ PQDs were synthesized using a colloidal synthesis method. The structural and optical properties of the resulting engineered PQDs were thoroughly characterized. These PQDs were then applied during the antisolvent step of perovskite film fabrication, enabling their simultaneous formation and integration at critical grain boundaries. PSCs incorporating these engineered PQDs exhibited significantly enhanced photovoltaic performance, characterized by increased PCE, suppressed hysteresis, and improved operational stability. These findings underscore the potential of in situ grown core–shell PQDs as an effective and scalable strategy for advancing the efficiency and durability of next-generation perovskite photovoltaic devices.

## 2. Materials and Methods

### 2.1. Preparation of Perovskite Nanoparticles

Methylammonium-tetraoctylammonium lead bromide core-shell nanoparticles were synthesized through a multi-step process to ensure high-quality formation. Initially, 0.16 mmol of methylammonium bromide (MABr, 80 wt%) and 0.2 mmol of lead(II) bromide (PbBr_2_) were dissolved in 5 mL of dimethylformamide (DMF) under continuous stirring. To this solution, 50 µL of oleylamine and 0.5 mL of oleic acid were added, forming the final core precursor solution. For the shell structure, a separate glass vial was used to dissolve 0.16 mmol of tetraoctylammonium bromide (t-OABr, 20 wt%) following the same protocol used for the core precursor solution. To facilitate nanoparticle growth, 5 mL of toluene was heated to 60 °C in an oil bath under continuous stirring. A 250 µL aliquot of the core precursor solution was rapidly injected into the heated toluene, initiating the formation of MAPbBr_3_ nanoparticles. Subsequently, a controlled amount of the t-OABr-PbBr3 precursor solution was injected into the reaction mixture, leading to the development of core-shell nanoparticles, as indicated by the emergence of a green color. The reaction was allowed to proceed for 5 min before the solution was transferred into a centrifuge tube for purification. The purification process involved centrifugation at 6000 rpm for 10 min, during which the precipitate was discarded and the supernatant was collected. To ensure further refinement, the supernatant was subjected to an additional centrifugation step with isopropanol at 15,000 rpm for 10 min. Finally, the resulting precipitate was redispersed in chlorobenzene, ensuring the stability of the nanoparticles for subsequent applications.

### 2.2. Preparation of Perovskite Solar Cell

Perovskite solar cells were fabricated on a transparent fluorine-doped tin oxide (FTO) substrate following a systematic process. Initially, the FTO substrates underwent a thorough cleaning procedure in an ultrasonic bath, starting with sonication in a soap solution, followed by sequential rinsing with distilled water, ethanol, and acetone. Afterward, the cleaned substrates were treated in a UV-ozone cleaner for 15 min to remove organic contaminants. The substrates were then preheated on a hot plate at 450 °C for 30 min. A compact titanium dioxide (TiO_2_) layer was deposited via spray pyrolysis, after which the substrates were maintained at 450 °C for an additional 30 min to ensure proper film formation. A mesoporous TiO_2_ layer was subsequently applied by spin-coating a colloidal dispersion of TiO_2_ paste (18NRT, Dyesol, Queanbeyan, Australia) in ethanol (1:6 ratio) at 4000 rpm for 30 s under ambient conditions, followed by annealing at 450 °C for 30 min. The perovskite precursor solution was prepared by dissolving 1.6 M PbI_2_, 1.51 M FAI, 0.04 M PbBr_2_, 0.33 M MACl, and 0.04 M MABr in 1 mL of a solvent mixture of dimethylformamide (DMF) and dimethyl sulfoxide (DMSO) in an 8:1 volume ratio. The perovskite film was then deposited using a two-step spin-coating process: 2000 rpm for 10 s, followed by 6000 rpm for 30 s. During the final 18 s of spinning, 200 µL of PVNPs, prepared in chlorobenzene (CB) at varying concentrations (3, 6, 9, 12, 15, 20, 25, 30 mg/mL), was introduced as an antisolvent. The films were subsequently annealed at 100 °C for 10 min and then at 150 °C for another 10 min to facilitate crystallization in a dry air atmosphere. Following perovskite film formation, Spiro-OMeTAD (Merck KGaA company, Darmstadt, Germany) was spin-coated as a hole transport layer at 4000 rpm for 20 s. Finally, an 80 nm thick gold (Au) counter electrode was deposited via thermal evaporation, completing the device fabrication process.

## 3. Results and Discussion

The PQDs utilized in this study were synthesized using a modified ligand-assisted reprecipitation (LARP) method, as detailed in the Appendix A and consistent with previously reported protocols [29,34]. To achieve the best structural and optical properties of the PQDs, we employed an optimized precursor molar ratio of MAPbBr_3_ to tetra-OAPbBr_3_ at 8:2 mixing ratio, as reported in the literature [29]. Comprehensive characterizations of the resulting MAPbBr_3_@tetra-OAPbBr_3_ PQDs were conducted to verify the successful formation of a uniform, passivated core–shell nanostructure. Transmission electron microscopy (TEM), X-ray diffraction (XRD), confocal laser scanning microscopy, and UV–vis spectroscopy were used to examine the morphological, structural, and optical properties of the synthesized nanocrystals. As shown in Figure 1a, the low-magnification TEM image reveals well-dispersed, square-shaped nanocrystals with an average size of approximately 15 nm, indicating excellent control over particle morphology and high colloidal stability. The high-resolution transmission electron microscopy (HRTEM) image presented in Figure 1b confirms the crystalline structure of the synthesized perovskite quantum dots (PQDs). The image reveals well-resolved lattice fringes with an interplanar spacing of approximately 2.9 Å, which corresponds to the (200) planes of the cubic MAPbBr_3_ phase. This d-spacing matches closely with that of the host perovskite film, indicating a high degree of crystallographic compatibility. Such lattice matching is essential for promoting epitaxial growth of the PQDs on the perovskite surface during the passivation process. The epitaxial interface is expected to minimize interfacial defects and facilitate efficient charge transfer, thereby enhancing the structural coherence and electronic coupling between the PQDs and the underlying perovskite matrix. Additionally, a slight contrast between the core and the surrounding region in Figure 1 (inset) suggests successful encapsulation by the TOAPbBr_3_ shell, which imparts steric protection and surface passivation, thereby enhancing environmental stability and reducing non-radiative surface recombination.

X-ray diffraction (XRD) analysis in Figure 1c shows three dominant diffraction peaks located at 2θ = 14.9°, 30.0°, and 45.0°, corresponding to the (100), (200), and (300) crystallographic planes of the cubic perovskite phase, respectively. The sharpness and high intensity of these peaks confirm the high crystallinity of the PQDs, while the absence of secondary phases or amorphous background signals indicates phase purity and preservation of the MAPbBr_3_ crystal structure despite the presence of the TOAPbBr_3_ shell. The XRD pattern of the perovskite quantum dots (PQDs) aged for three months, as shown in Figure 1c, demonstrates excellent structural stability over time. The preservation of distinct diffraction peaks corresponding to the cubic MAPbBr_3_ phase indicates that the PQDs retain their crystallinity and phase integrity without significant degradation. This long-term stability is a critical attribute for their application as a passivation layer in perovskite solar cells (PSCs), where consistent structural and chemical properties are essential to enhance device longevity. Figure 1d illustrates the optical characteristics of the PQDs through both absorbance and photoluminescence (PL) spectra. The absorbance curve exhibits a steep onset near 510 nm, indicative of a well-defined band edge and strong quantum confinement effects, consistent with the nanoscale size of the crystals. The broad absorption tail extending into the blue and near-UV regions reflects the high absorption coefficient typical of halide perovskites and facilitates strong light-harvesting capabilities. Notably, the absence of sub-bandgap features in the absorption profile suggests minimal trap-state density, further implying effective surface passivation by the TOAPbBr_3_ shell. The PL spectrum shows a narrow and intense emission peak centered at approximately 525 nm, with a full width at half maximum (FWHM) of ~20 nm, which is characteristic of high-quality PQDs with minimal non-radiative recombination. The small Stokes shift (~15 nm) between the absorption edge and emission maximum indicates efficient exciton confinement, reducing self-absorption losses by minimizing spectral overlap between absorption and emission.

Building on the designed structural and optical characteristics of the synthesized MAPbBr_3_@tetra-OAPbBr_3_ PQDs, we explored their role as surface passivation agent in PSCs. To evaluate their impact, we fabricated incomplete structured PSCs with the following architecture: an FTO glass/compact TiO_2_/perovskite/PQD passivation layer. The PQDs were introduced atop the perovskite active layer via an in situ passivation strategy integrated into the antisolvent deposition step, as illustrated in Figure 2 and described in detail in the Materials and Methods section. During the antisolvent-assisted crystallization step, commonly used to promote rapid nucleation and film formation in perovskite solar cells, perovskite PQDs dispersed in the antisolvent are co-deposited onto the growing perovskite film. This approach is expected to enable the uniform distribution of PQDs along grain boundaries and across the surface of the perovskite layer.

To investigate the morphological effects of PQD incorporation, scanning electron microscopy (SEM) was performed, as shown in Figure 3. SEM images of perovskite films prepared with different PQD concentrations (0, 3, 6, 9, 12, 15, 20, and 30 mg/mL) have been provided. PQDs were added to the chlorobenzene (CB) antisolvent used during crystallization, allowing the NH_3_Br functional groups of PQDs to interact effectively with the perovskite precursors, consistent with previously reported methods for enhancing crystallization and passivation in perovskite solar cells [35,36]. Macroscopic observations showed clear variations in film uniformity and coverage as PQD concentrations changed, with enhanced film quality particularly noticeable at intermediate PQD concentrations. Microscopic analysis via SEM revealed that films without PQD exhibited higher densities of grain boundaries. Increasing PQD concentrations up to 15 mg/mL reduced grain boundary density and improved overall film uniformity, confirming effective passivation and enhanced crystallinity facilitated by the NH_3_Br functional groups. At the optimized PQD concentration, high-magnification SEM images (Appendix A) reveal that the PQDs were predominantly located at the grain boundaries of the perovskite film, suggesting their role in passivating interfacial defects and enhancing film morphology. Morphological improvement occurred at a PQD concentration of 15 mg/mL, correlating with the highest device efficiency and stability. However, at concentrations above 15 mg/mL, slight aggregation and non-uniformity appeared, aligning with the observed decrease in photovoltaic performance at higher PQD loadings. These results reinforce the crucial role of optimized PQD concentrations in enhancing perovskite film morphology and performance through effective passivation of surface defects and grain boundaries.

We explicitly compared our in situ PQD passivation strategy with previous approaches employing surface-coordinating molecules and nanostructures, specifically those discussed in references [37,38]. Prior studies have demonstrated that terpyridine-zinc(II) coordination nanosheets act as heterogeneous nucleation seeds, effectively enhancing crystallization and defect passivation through surface chemical interactions. In contrast, our approach involves in situ epitaxial growth of core–shell structured perovskite quantum dots (PQDs), composed of MAPbBr_3_ cores and tetraoctylammonium PbBr_3_ shells, incorporated during the antisolvent-assisted crystallization step. We propose the following mechanism for PQD-induced passivation: the tetraoctylammonium PbBr_3_ shell surrounding the PQD core provides robust chemical and physical protection. The Br-rich shell effectively interacts with undercoordinated Pb^2+^ ions and halide vacancies, prevalent at grain boundaries, significantly reducing non-radiative recombination losses. Moreover, the literature confirms the enrichment of bromide ions at grain boundaries in iodide-based perovskite compositions [39,40], supporting our expectation that the bromide-rich tetraoctylammonium PbBr_3_ shells efficiently passivate grain boundary defects.

The photovoltaic characteristics of the fabricated perovskite solar cells (PSCs), utilizing a standard n–i–p architecture (FTO/c-TiO_2_/m-TiO_2_/perovskite/PQDs/spiro-OMeTAD/Au), were significantly enhanced by incorporating perovskite quantum dots (PQDs) during the antisolvent dripping step of the perovskite film formation. Prior to their addition, dynamic light scattering (DLS) analysis was conducted to assess the colloidal stability of the PQD dispersion at different concentrations. As shown in Appendix A, the results confirmed a uniform particle size distribution and the absence of significant agglomeration, ensuring homogenous PQD incorporation.

To statistically validate the effect of PQD passivation, photovoltaic measurements were conducted on 10 independently fabricated PSC devices for each PQD concentration. As depicted in Figure 4a–c, the PQD-treated devices consistently outperformed the pristine controls. The optimal PQD concentration led to marked enhancements in short-circuit current density (Jsc), open-circuit voltage (Voc), fill factor (FF), and overall power conversion efficiency (PCE). The average photovoltaic parameters and standard deviations, derived from 10 devices per condition, are summarized in the Table 1 below, confirming both the performance benefits and excellent reproducibility of the PQD-assisted passivation approach.

The device performance steadily increased with PQD addition up to 15 mg/mL, beyond which a gradual decline occurred. This trend suggests that while moderate PQD content improves film quality and electronic properties, excessive loading could disrupt charge transport or introduce new recombination pathways. The enhancement in Voc (from 1.120 V to 1.137 V) and PL quenching observed in Figure 4d reflect the critical role of PQDs in defect passivation. During the antisolvation step, the PQDs, having similar composition and lattice structure to the host perovskite phase, embed themselves at the grain boundaries and surface terminations of the perovskite layer. These boundaries are typically rich in halide vacancies, under-coordinated lead ions (Pb^2+^), and other trap sites that act as non-radiative recombination centers. The PQDs interact with these trap states via favorable ion exchange and interfacial bonding, especially through halide migration compensation and Pb–halide coordination, effectively “healing” these defects.

Additionally, the small size and high surface-to-volume ratio of PQDs provide an extensive surface for passivation interactions and contribute to grain size enlargement and film densification. This is corroborated by the improvement in FF (from 70.1% to 77%) and Jsc (from 24.5 to 26.1 mA/cm^2^), indicating more efficient charge collection and suppressed interfacial losses. The improved film morphology, facilitated by PQDs, also enhances charge transport pathways across the active layer and reduces series resistance. However, at concentrations ≥15 mg/mL, performance degradation is observed. While the observed steady-state photoluminescence (PL) quenching in PQD-treated perovskite films indicates enhanced interfacial interactions, it alone does not definitively confirm improved charge extraction or suppressed non-radiative recombination. Time-resolved photoluminescence (TRPL) or transient absorption spectroscopy (TAS) would offer more direct insight into charge carrier lifetimes and recombination dynamics. However, due to current instrumental limitations, such measurements were not feasible within the scope of this study. Nonetheless, the conclusion of reduced recombination is supported by the concurrent improvements in open-circuit voltage (Voc), fill factor (FF), and power conversion efficiency (PCE) across 10 independently fabricated devices with high reproducibility. These performance enhancements, together with the PL quenching, strongly suggest that the PQD passivation strategy improves interfacial charge transfer and suppresses non-radiative pathways.

The improved optoelectronic response and operational durability of PSCs fabricated with PQDs, as demonstrated in Figure 5, can be realistically attributed to a combination of physical and chemical passivation mechanisms. The incident photon-to-current efficiency (IPCE) spectra in Figure 5a show that the PQD-treated device exhibits consistently higher quantum efficiency across the visible spectrum, especially in the 400–750 nm region, compared to the pristine device. This enhancement is a strong indication of reduced interfacial and bulk recombination, which is likely facilitated by the role of PQDs in passivating defects and improving film uniformity.

During the antisolvation step in film fabrication, the addition of PQDs may promote localized improvements in crystallization and morphology by acting as nucleation sites or modifying the perovskite growth kinetics. The interaction and passivation mechanism, described as ion exchange and Pb–halide coordination, is experimentally supported by our SEM analyses and correlated photovoltaic performance metrics. SEM images of perovskite films prepared with varying PQD concentrations clearly indicate morphological improvements with PQD addition. Specifically, films without PQDs exhibited higher densities of grain boundaries, indicative of numerous defect sites. However, increasing PQD concentrations up to an optimal level (15 mg/mL) significantly reduced grain boundary density and enhanced overall film uniformity, supporting effective defect passivation facilitated by PQDs through the interaction of NH_3_Br functional groups. Beyond this optimal concentration, slight aggregation and non-uniformities appeared, aligning with diminished photovoltaic performance. This morphological evidence from SEM, combined with corresponding improvements in photovoltaic parameters such as enhanced Voc, Jsc, and fill factor, collectively supports the efficacy of our defect-healing mechanism involving PQD interactions at grain boundaries and surfaces. Additionally, incident photon-to-current efficiency (IPCE) spectra revealed enhanced carrier extraction across the visible spectrum, indicating reduced defect-related recombination. TEM analysis confirmed PQD integration and stable core–shell structure. Furthermore, X-ray diffraction (XRD) measurements affirmed phase purity and structural stability, supporting the assertion that PQDs supply necessary ions (e.g., Br^−^) without compromising their structural integrity. Collectively, these findings substantiate our proposed mechanism involving PQD interactions and defect passivation at grain boundaries and surfaces.

In terms of device stability, as demonstrated in Figure 5b, the PQD-containing devices retained more than 92% of their initial PCE after 900 h under ambient conditions, while pristine devices degraded to about 80%. This realistic improvement in long-term performance suggests that PQDs contribute to localized stabilization rather than acting as a complete moisture barrier. PQDs may help mitigate ion migration (especially of I^−^ and MA^+^) by reducing the number of mobile halide defects and strengthening grain boundary interfaces, where degradation often initiates. Additionally, the slightly denser morphology induced by PQDs can limit the diffusion pathways for oxygen and water, thus slowing chemical degradation processes such as the formation of PbI_2_ or the loss of organic cations.

It is also plausible that PQDs slightly modulate the local electric field distribution at the interface between the perovskite and hole transport layer, reducing interfacial instability and enhancing charge extraction over time. However, the passivation effect is highly dependent on PQD loading: excessive concentrations may result in phase separation or trap-rich interfaces, which can negate these benefits. The modest but consistent enhancement in both IPCE and operational stability observed in the PQD-treated devices suggests that the optimized PQD incorporation enables a balanced passivation strategy, effectively reducing recombination without disrupting the electronic landscape of the device.

In addition to device performance improvements, the practical scalability of the PQD incorporation strategy is an essential consideration for advancing perovskite solar cell (PSC) manufacturing. Although the use of spin-coating combined with a PQD-containing antisolvent has demonstrated clear benefits at the laboratory scale, we acknowledge that this approach poses challenges for large-area module fabrication and roll-to-roll production. These challenges include difficulties in maintaining uniform film coverage, material efficiency, and process consistency across extended surfaces.

To overcome these limitations, our PQD integration strategy can be readily adapted to scalable deposition techniques such as blade coating or slot-die printing. These techniques enable precise control over film thickness, uniformity, and deposition rates, making them well suited to industrial-scale fabrication. For example, PQDs may be directly mixed into the perovskite precursor solution and applied via scalable methods, as previously demonstrated [41]. Alternatively, sequential deposition methods, also compatible with scalable processing, allow for PQD incorporation during the second step of film formation. Notably, this step often utilizes isopropanol as a solvent [42], and we have verified that our synthesized PQDs are well dispersed in isopropanol without aggregation. These findings highlight the feasibility of implementing our PQD-assisted passivation strategy within scalable manufacturing platforms for high-performance, large-area perovskite solar modules.

## 4. Conclusions

In this study, we successfully synthesized PQDs with a MAPbBr_3_@TOAPbBr_3_ core–shell architecture, exhibiting excellent structural integrity and optical properties tailored for epitaxial passivation in PSCs. These PQDs were strategically incorporated onto the perovskite active layer during the antisolvent-assisted crystallization process, enabling intrinsic passivation of surface and grain-boundary defects. PSC devices treated with an optimized PQD concentration of 15 mg/mL demonstrated a notable enhancement in photovoltaic performance, achieving a PCE of 22.85%, a Jsc of 26.1 mA/cm^2^, Voc of 1.137 V, and FF of 77%, all of which significantly surpassed those of pristine devices. IPCE spectra further confirmed improved charge collection efficiency across the visible range. Moreover, long-term stability assessments revealed more than 92% PCE retention after 900 h of ambient storage, in contrast to approximately 80% for untreated devices. These findings establish PQDs as effective nanoscale passivating agents that not only mitigate non-radiative recombination but also enhance the operational stability of PSCs, all while maintaining compatibility with standard fabrication protocols.

## Figures and Tables

**Figure 1 nanomaterials-15-00978-f001:**
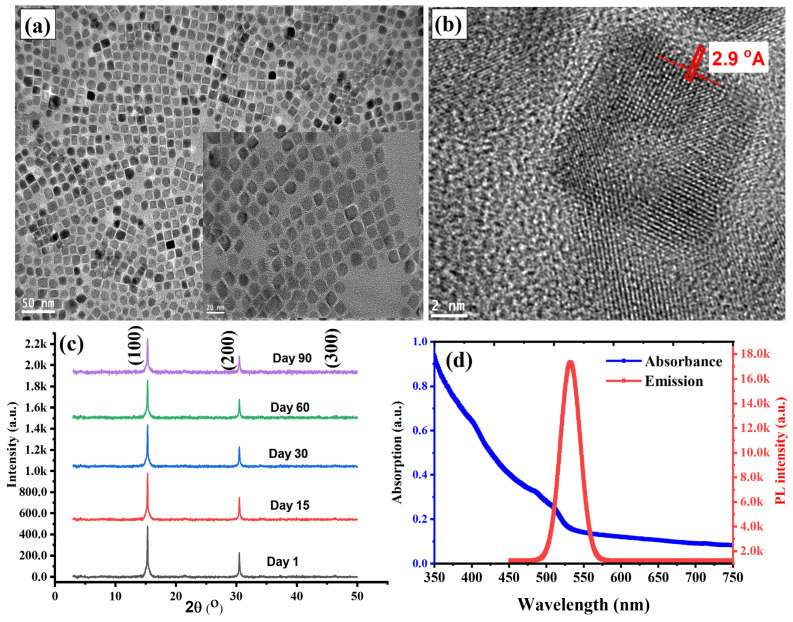
Structural and optical characterization of the synthesized MAPbBr_3_@tetra-OAPbBr_3_ perovskite quantum dots (PQDs). (**a**) Low-magnification TEM image showing well-dispersed, square-shaped nanocrystals with an average size of ~15 nm, indicating uniform morphology and excellent colloidal stability. The inset highlights a slight contrast between the core and shell, indicating successful passivation by the tetra-OAPbBr_3_ shell. (**b**) High-resolution TEM image revealing well-defined lattice fringes with an interplanar spacing of 2.9 Å, corresponding to the (200) planes of cubic MAPbBr_3_. This lattice spacing matches that of the host perovskite matrix, supporting epitaxial growth and strong interfacial coupling. (**c**) XRD patterns of fresh and aged PQDs showing distinct diffraction peaks at 2θ = 14.9°, 30.0°, and 45.0°, assigned to the (100), (200), and (300) planes of cubic perovskite. The absence of secondary phases and the retention of peak sharpness after three months confirm high crystallinity and excellent phase stability. (**d**) Absorption and photoluminescence (PL) spectra showing a sharp absorption onset near 510 nm and a narrow, intense PL peak centered at ~525 nm. The steep absorbance edge and absence of sub-bandgap features suggest strong quantum confinement and effective surface passivation by the TOAPbBr_3_ shell.

**Figure 2 nanomaterials-15-00978-f002:**
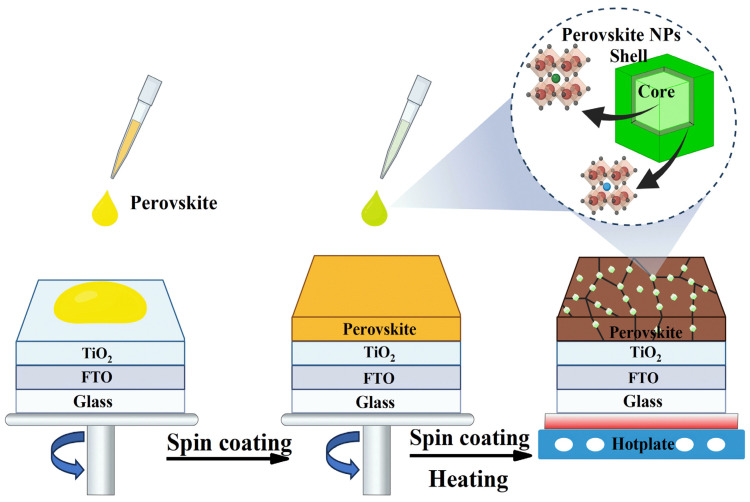
Schematic illustration of the in situ deposition of PQDs during the antisolvent-assisted crystallization step, enabling their conformal integration at grain boundaries and surfaces of the perovskite film.

**Figure 3 nanomaterials-15-00978-f003:**
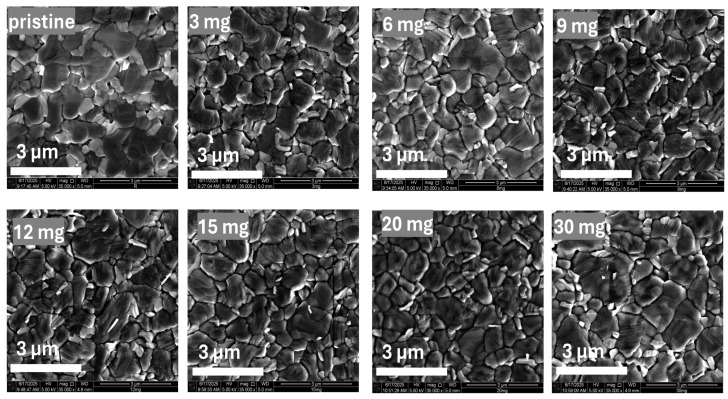
Scanning electron microscopy (SEM) images of perovskite films prepared with varying concentrations of perovskite quantum dots (PQDs) added to the chlorobenzene (CB) antisolvent during film crystallization (0, 3, 6, 9, 12, 15, 20, and 30 mg/mL). Morphological analysis reveals a progressive reduction in grain boundary density and improvement in film uniformity with increasing PQD concentration, reaching optimal morphology at 15 mg/mL. At higher loadings (>15 mg/mL), aggregation and non-uniform features emerge, correlating with the decline in device performance.

**Figure 4 nanomaterials-15-00978-f004:**
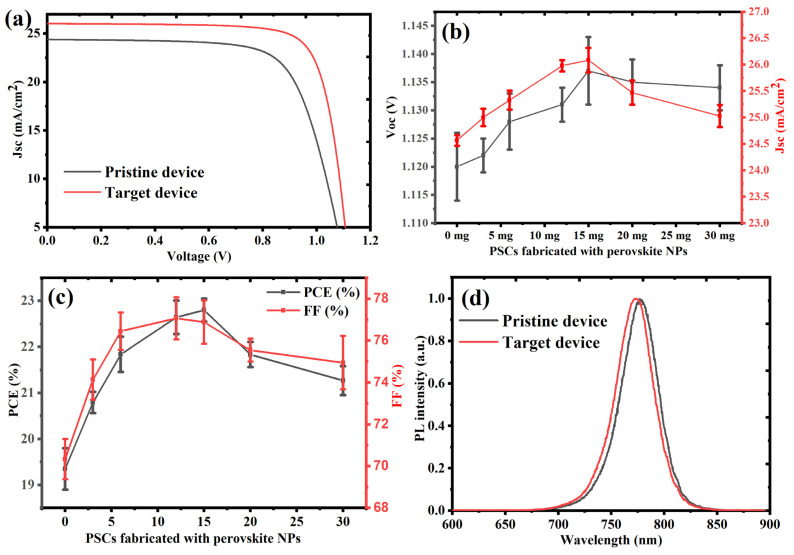
(**a**) J–V characteristics of pristine and PQD-treated perovskite solar cells (PSCs) measured under standard AM 1.5G illumination, illustrating enhanced Jsc and Voc with PQD incorporation. (**b**) Statistical variation of short-circuit current density (Jsc) and open-circuit voltage (Voc) with increasing PQD concentration, based on measurements from 10 devices per condition, revealing peak values at 15 mg/mL. (**c**) Average power conversion efficiency (PCE) and fill factor (FF) as functions of PQD concentration, also derived from 10-device datasets, demonstrating optimal photovoltaic performance at 15 mg/mL. (**d**) Steady-state photoluminescence spectra of pristine and PQD-treated perovskite films, where suppressed PL intensity in treated samples indicates improved defect passivation and enhanced charge extraction.

**Figure 5 nanomaterials-15-00978-f005:**
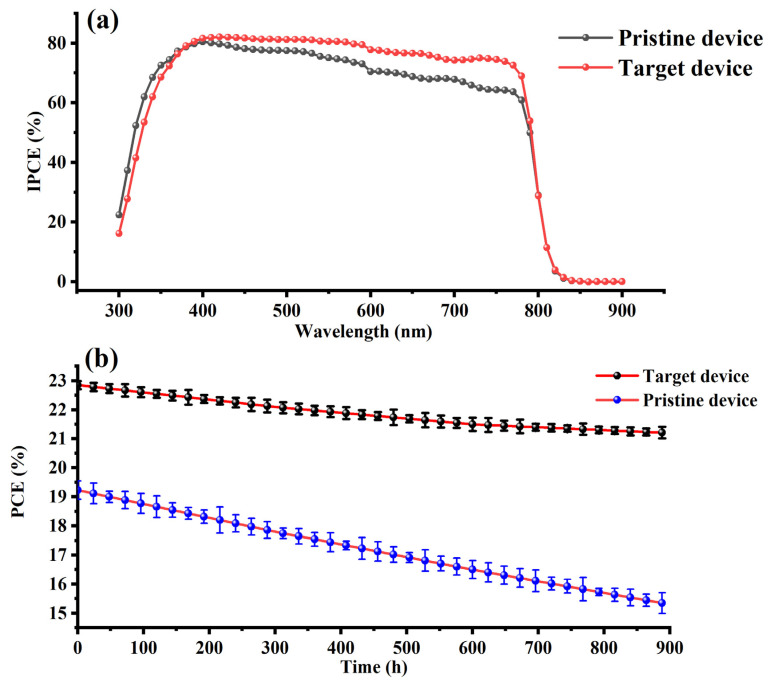
(**a**) Incident photon-to-current efficiency (IPCE) spectra of pristine and PQD-treated perovskite solar cells (PSCs). The target device exhibits a higher and broader photo response across the visible spectrum (300–800 nm), indicating enhanced light harvesting and reduced recombination due to effective passivation by PQDs. (**b**) Long-term ambient stability of the pristine and PQD-treated PSCs stored under controlled conditions (25 °C, ~35% RH, dark). The PQD-based device retained more than 92% of its initial power conversion efficiency (PCE) after 900 h, compared to ~80% retention in the pristine device, highlighting the role of PQDs in suppressing degradation and improving device robustness.

**Table 1 nanomaterials-15-00978-t001:** A summary of photovoltaic parameters (Jsc, Voc, FF, and PCE) measured across 10 perovskite solar cell (PSC) devices incorporating varying concentrations of perovskite quantum dots (PQDs) during the anti-solvent treatment step.

PSC Devices with PQDs (mg/mL)	PCE (%)	Voc (V)	Jsc (mA/cm^2^)	FF (%)
0	19.35 ± 0.45	1.120 ± 0.006	24.56 ± 0.10	70.33 ± 0.96
3	20.79 ± 0.23	1.122 ± 0.003	25.00 ± 0.16	74.14 ± 0.96
6	21.84 ± 0.38	1.128 ± 0.005	25.33 ± 0.18	76.45 ± 0.89
12	22.64 ± 0.37	1.131 ± 0.003	25.98 ± 0.11	77.07 ± 1.01
15	22.80 ± 0.25	1.137 ± 0.006	26.08 ± 0.23	76.88 ± 1.04
20	21.84 ± 0.27	1.135 ± 0.004	25.47 ± 0.23	75.55 ± 0.54
30	21.27 ± 0.32	1.134 ± 0.004	25.03 ± 0.21	74.95 ± 1.27

## Data Availability

The original contributions presented in this study are included in the article. Further inquiries can be directed to the corresponding author.

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
