# Peer review of "In Situ Epitaxial Quantum Dot Passivation Enables Highly Efficient and Stable Perovskite Solar Cells"

_nanomaterials, 2025, doi:10.3390/nano15130978_

Round 1

Reviewer 1 Report

Comments and Suggestions for Authors

Reviewer Comments:

The authors report a dual-functional passivation strategy for perovskite solar cells (PSCs) using in situ epitaxially integrated core–shell perovskite quantum dots (MAPbBr₃@TOAPbBr₃ PQDs). The manuscript is well-organized and presents promising photovoltaic performance (PCE of 22.85%, Voc of 1.137 V, Jsc of 26.1 mA/cm², FF of 77%) as well as notable long-term stability. However, several critical aspects require clarification or deeper analysis before the manuscript can be considered for publication. I raise the following points for the authors to address:

  • Clarification of Core–Shell Structural Stability:
    The stability of PQDs in ambient air is highlighted as a key factor behind improved long-term device performance. However, more experimental evidence is needed to confirm the robustness of the core–shell structure under environmental stress (humidity, temperature). Have the authors performed XRD or TEM on aged PQDs? Such data would reinforce the claim that the shell structure effectively protects the core and maintains passivation efficacy.

  • Mechanism of Epitaxial Compatibility and Passivation:
    The authors mention epitaxial compatibility between MAPbBr₃ PQDs and the host perovskite. However, no lattice matching data or crystallographic orientation relationships are shown. Have the authors performed high-resolution TEM (HRTEM) or selected area electron diffraction (SAED) to validate this claim? Additional crystallographic evidence would solidify the proposed mechanism of epitaxial passivation.

  • Charge Carrier Dynamics Missing:
    To better understand the reduced non-radiative recombination, time-resolved photoluminescence (TRPL) or transient absorption spectroscopy (TAS) measurements should be provided. The observed PL quenching alone is insufficient to deduce improved carrier extraction or lifetime extension. Including TRPL decay times would strengthen the discussion on charge dynamics.

  • Insufficient Comparison with Passivation Strategies:
    The concept of perovskite crystallization and defect passivation has been reported previously. To enhance the scientific context, the authors should compare their in situ PQD strategy with prior works that utilized surface-coordinating molecules, particularly: 10.1021/acsami.5c05011, 10.1039/D3TA00505D and 10.1039/D3TA00505D. These studies demonstrated that coordination ligands or seeded nanocrystal growth can regulate perovskite film crystallinity and passivate surface defects. Please elaborate on how your PQD approach differs mechanistically or performs comparatively better in efficiency or stability.

  • Lack of Device Reproducibility and Statistical Analysis:
    The device metrics are only presented as single values. To confirm the reliability of the results, statistical analysis of device performance should be included. Please provide box plots or histograms showing the PCE distribution over multiple devices (e.g., n = 10–20), along with mean ± standard deviation for each key parameter (Voc, Jsc, FF, PCE).

  • Overloaded Antisolvent Step—Potential Scalability Issues:
    The method relies on dispersing PQDs in the antisolvent and applying them during spin-coating. While effective in lab-scale devices, it is unclear how this step would scale for larger-area modules or roll-to-roll processing. Could the authors comment on the process compatibility and whether PQDs can be integrated using scalable deposition techniques like blade coating or slot-die printing?

Author Response

Dear reviewer,

Best regards,

Masfer

Reviewer 2 Report

Comments and Suggestions for Authors

This manuscript reports an effective in situ epitaxial core–shell perovskite quantum dot passivation strategy that improves perovskite solar cell efficiency and stability. The work is well conducted and significant, but some points require clarification before acceptance.

  1. The introduction should be updated to mention that certified PSC efficiencies now exceed 27%.
  2. Please include DLS data for PQDs at different concentrations to check for aggregation.
  3. Provide macroscopic photos and microscopic images (surface and cross) of films with varying PQD concentrations to show effects on morphology.
  4. Please clarify and provide evidence that PQDs reside at grain boundaries rather than on surfaces, buried interfaces, or as aggregates.
  5. Since PQDs are capped with organic ligands, a discussion or data on whether these ligands impede or facilitate charge carrier transport is necessary.

Author Response

Dear reviewer,

Best regards,

Masfer

Reviewer 3 Report

Comments and Suggestions for Authors

The manuscript explores the potential for enhancing the efficiency and stability of perovskite solar cells by incorporating perovskite core-shell nanoparticles into the perovskite active layer before crystallization. The improvement in solar cells' efficiency (higher Voc, photocurrent, and fill factor, with better stability) was attributed to the passivation of defects on the grain boundaries and surface.

The subject is interesting from a scientific and applied point of view.

The experiment is described in detail, the discussion sounds plausible, manuscript is well organized.

However, before my recommendation for publication, I have a few remarks.  

Are the nanoparticles homogeneously dispersed throughout the material or do they form a thin film on the surface of the active perovskite layer?

186 „The large Stokes shift between the absorption edge and emission maximum may also indicate efficient exciton confinement and reduced self-absorption losses.“

Is it possible that the shift of the luminescent peak is a consequence of the presence of nanoparticles that have a larger optical gap?

231 „These boundaries are typically rich  in halide vacancies, under-coordinated lead ions (Pb²⁺), and other trap sites that act as  non-radiative recombination centres. The PQDs interact with these trap states via favourable ion exchange and interfacial bonding, especially through halide migration compensation and Pb–halide coordination, effectively "healing" these defects.“

What is proof of this model? Besides, if it is correct, the annihilation of the defect-vacancy in the active perovskite material results in the creation of a defect (vacancy) in the nanoparticle.

Author Response

Dear reviewer,

Best regards,

Masfer

Round 2

Reviewer 1 Report

Comments and Suggestions for Authors

The revision has been completed to a publishable level